# Mapping *Staphylococcus aureus* at Early and Late Stages of Infection in a Clinically Representative Hip Prosthetic Joint Infection Rat Model

**DOI:** 10.3390/microorganisms12091895

**Published:** 2024-09-14

**Authors:** Mariam Taha, Abdullah AlDuwaisan, Manijeh Daneshmand, Mazen M. Ibrahim, Jonathan Bourget-Murray, George Grammatopoulos, Simon Garceau, Hesham Abdelbary

**Affiliations:** 1Chronic Disease Program, The Ottawa Hospital Research Institute, Ottawa, ON K1H 8L6, Canada; 2Division of Orthopaedic Surgery, The Ottawa Hospital, Ottawa, ON K1H 8L6, Canada; aalduwaisan92@gmail.com (A.A.); mibra020@uottawa.ca (M.M.I.); jbourgetmurray@gmail.com (J.B.-M.); ggrammatopoulos@toh.ca (G.G.); sigarceau@toh.ca (S.G.); 3Division of Orthopaedic Surgery, Faculty of Medicine, Kuwait University, Jabriya, Kuwait; 4Department of Pathology and Laboratory Medicine, University of Ottawa, Ottawa, ON K1H 8M5, Canada; mj.dan18@outlook.com

**Keywords:** bacteria, animal model, histologic analysis, periprosthetic joint infection, *Staphylococcus aureus*, titanium, 3D printed implant, hip surgery, cemented hip hemiarthroplasty

## Abstract

Prosthetic joint infection (PJI) continues to be a devastating complication following total joint replacement surgeries where *Staphylococcus aureus* is the main offending organism. To improve our understanding of the disease pathogenesis, a histological analysis of infected peri-implant tissue in a hip PJI rat model was utilized to assess *S. aureus* spread and tissue reaction at early and late stages of infection. Sprague–Dawley rats were used and received a left cemented hip hemiarthroplasty using a 3D-printed titanium femoral stem. The rats received an intra-articular injection of *S. aureus* Xen36. These infected rats were sacrificed either at 3 days post-infection (early-stage infection) or at 13-days post-infection (late-stage infection). The femoral and acetabular tissues of all animals were harvested at euthanasia. Histological analysis for the harvested tissue was performed using immunohistochemistry, hematoxylin and eosin, as well as Masson’s trichrome stains. Histological examination revealed significant quantitative and qualitative differences in peri-implant tissue response to infection at early and late stages. This hip PJI rat model identified clear histologic differences between early and late stages of *S. aureus* infection and how quickly bacterial infiltration could occur. These findings can provide insight into why certain surgical strategies like debridement and antibiotics may be associated with high failure rates.

## 1. Introduction

Prosthetic joint infection (PJI) continues to be the leading cause of reoperation, prolonged hospital admissions, and mortality despite advances in infection prevention protocols [1,2]. The pooled incidence for periprosthetic infection following hip hemiarthroplasty (HA) is estimated to be ~3% [3,4,5]. Current treatment strategies of PJI involve debridement, antibiotic therapy, and implant retention (DAIR). However, this approach has a failure rate reaching 36% [6,7,8,9] including significant morbidity and may result in limb amputations or even death [10]. Systemic antibiotic therapies often fail because they are unable to reach minimum inhibitory concentrations (MICs) at the site(s) of infection due to various reasons. One of them is biofilm formation on the implant surface and surrounding tissue, shielding resident bacteria and leading to antibiotic tolerance [11,12]. *Staphylococcus aureus* is the most common pathogen isolated from PJI patients, being present in 30–40% of all PJI cultures. *S. aureus* is capable of forming biofilm on implant surface and the surrounding tissue in PJI settings. Moreover, reports demonstrated the capacity of *S. aureus* to infiltrate into the canalicular network of bone structure and survive within the host osteoblast and immune cells [13,14]. All these factors can contribute to the failure of DAIR treatment and the reoccurrence of the infection.

Developing a clinically representative PJI animal model is a critical step towards improving our understanding of disease pathogenesis and validating the efficacy of our current treatment strategies. The hip hemiarthroplasty (HA) PJI rat model described by Hadden et al. is currently the most clinically representative HA-PJI model due to its unique features of cement fixation, “ball-in-socket” mechanics, implant design biofilm formation on implant surface, and ability to weight-bear while demonstrating an antalgic gait associated with infection [15,16].

Histological analysis is needed to further validate this model’s PJI disease process and determine if it can be reliably used to assess novel therapeutic interventions and strategies to improve HA-PJI outcomes [17,18,19,20]. Using this established *S. aureus* cemented hip HA-PJI rat model, the objectives for the study were (I) to map the pattern of bacterial spread, and (II) identify differences in tissue response over time in infected peri-implant tissues at early and late time points post-infection to identify differences in tissue response over time, and assess alterations in peri-implant tissue reaction to infection.

## 2. Materials and Methods

Approval for this study, including the use and handling of experimental animals, was granted by the University of Ottawa Animal Research Ethics Committee protocol OHRI 3988.

### 2.1. In Vitro Preparation of S. aureus

*S. aureus* Xen36 (ATCC 49525), a biofilm-forming, kanamycin-resistant, methicillin-sensitive, bioluminescent strain, was used as the infective agent [19,21,22]. From frozen stock, Xen36 was streaked into tryptic soy agar (TSA) plate and incubated overnight at 37 °C. Isolated colonies were used to inoculate tryptic soy broth (TSB). The suspension was incubated at 37 °C and 200 rpm. After the overnight incubation (16–17 h), 5 mL of the suspension was centrifuged at 3000× *g* for 10 min. The supernatant was removed, and the pellet was then resuspended in 1 mL 0.9% sterile saline. Xen36 concentration (~10^10^ CFU/mL) was confirmed by serial dilutions then plating it into TSA [15].

### 2.2. Cemented Hip HA PJI Animal Model

During hip HA surgeries the femur head is removed, and a prosthesis (metallic stem with a head) is inserted inside the femur canal whereas the head fits inside the socket (acetabulum). The stem prosthesis is fixed in the bone using either cemented or uncemented approach, depending on the quality of bone growth around the prosthesis. Here, we used the cemented approach for our HA surgeries. A total of nine Sprague–Dawley rats, weighing between 500 and 700 g, at 12 weeks of age were used. Animals were housed at University of Ottawa, Canada, animal care facility in identical conditions in individual cages. As previously described, briefly, all animals underwent left hind limb cemented HA via the posterior approach under general anesthesia with isoflurane [15]. A 3D-printed medical-grade titanium HA was implanted and cemented with non-antibiotic-loaded poly(methyl methacrylate) (PMMA) cement (Palacos medium viscosity; Heraeus, Hanau, Germany). The model implant was a monoblock manufactured in medical-grade porous Ti6Al4V [15,16]

Six rats received an intra-articular injection of 20 μL of *S. aureus* Xen36 (~2 × 10^8^ CFUs) after implantation and prior to wound closure. To map histological differences in the pattern of bacterial spread over time, three infected rats were euthanized at an early time point, postoperative day 3 (POD 3), and three infected animals were sacrificed at a later time point, POD 13 (Appendix A). The early time point was chosen based on our previous study and other studies which demonstrated that the in vivo bioluminescent signal peaked between 3 and 5 days post-infection [15,19,23,24]. Therefore, we wanted to assess histologic changes at this early peak in the infection process. The late time point of POD 13 was chosen due to the aggressive clinical damage that the infection caused to the animals’ health beyond that point [15]. As per Hadden et al., a later time point was associated with higher rates of dislocation and wound complications [15]. Moreover, other studies for their murine model used a similar time point to study chronic *S. aureus* infection [19,23]. Three non-infected rats that underwent HA surgery were euthanized on POD 13 as controls (Appendix A).

### 2.3. In Vivo Luminescent Imaging

Rats were anaesthetized with isoflurane and monitored with oxygen while in vivo bioluminescence imaging was captured using in vivo imaging system (IVIS) spectrum optical imager (Caliper Life Sciences, PerkinElmer, Waltham, MA, USA) on POD 1, 3, and 11 [15,25]. Data were acquired at f/8, medium binning, and automatic exposure time with a −90° detector with a background threshold of 600 photons/pixel at a maximal detection range of 400 nm to 900 nm. High-resolution images were created with color scale overlaid on a monochrome photograph of rats and quantified as maximum flux (radiance photons/s/cm^2^) within a circular region of interest (ROI) using Living Image software v. 4.7.4 (PerkinElmer) [15].

### 2.4. Bacterial Load in Tissue

Periarticular soft tissue abscess was harvested and processed for bacterial count. Collected femur and acetabulum were checked for bacterial load as well (Appendix A). Briefly, the abscess was ground, added to 1 mL of sterile saline, vortexed, and then passed through a 70 µm strainer. The head of the implant in the femur and the acetabulum socket were each washed with 1 mL sterile saline, then passed through a 70 µm strainer. The collected liquid was plated on TSA plates supplemented with 200 µg/mL kanamycin to select for *S. aureus* Xen36 (kanamycin-resistant). Plates were incubated overnight at 37 °C. Cultures were verified for Xen36 with IVIS [15].

### 2.5. Histology Analysis

#### 2.5.1. Fixation and Processing of Femur and Acetabulum

The left-sided femurs and acetabula of all rats were harvested and fixed in 10% buffered formalin (Fisher Scientific, Toronto, ON, Canada) for 14 [23,26,27] days then were added to 70% ethanol. The tissue was decalcified for 48 h in 12.5% formic acid. After decalcification, the implants were carefully removed from the femoral canal (Appendix A). The cement mantle always came out with the implant in one segment (Appendix A). All decalcified femurs and acetabula were then cut in half with a 15 blade at the same anatomic. The level of these cuts was 1 cm from the greater trochanter for the femur, and 1.5 cm from the iliac crest for the acetabulum. Each side of the cut specimen was painted with toluidine blue dye for orientation (Appendix A). Each cut specimen was embedded in the paraffin block with the same orientation (Appendix A). There was a total of 18 paraffin blocks for the femur and 18 paraffin blocks for the acetabulum. Each block was sectioned using Leica BondTM system at 5 µm thickness starting at the level of the blue dyed surface [23]. Two sections were taken per block (Appendix A) for each of the hematoxylin and eosin (H&E), Masson trichrome (MT), and immunohistochemistry stains [20]. H&E and MT were used to assess polymorphonuclear cell infiltration (PMN), as well as tissue necrosis and fibrosis, respectively, while IHC staining was utilized to localize *S. aureus* within the harvested peri-implant femoral and acetabular tissues.

#### 2.5.2. IHC Staining

Rabbit polyclonal *S. aureus* antibody (Abcam cat#ab20920, Cambridge, UK) was used to localize bacterial presence within femur and acetabulum tissue. IHC staining was performed on formalin fixed paraffin embedded tissue sections using the Leica Bond’M system. IHC sections were stained with sodium citrate buffer (pH 6.0, epitope retrieval solution1) for 20 min. The sections were then incubated using 1:400 dilution for 30 min at room temperature and detected using HRP conjugated compact polymer system. Slides were stained using DAB as the chromogen, counterstained with hematoxylin, mounted, and cover-slipped.

#### 2.5.3. Analysis of Histology Slides

All slides were assessed by an expert musculoskeletal pathologist. The entire tissue slices were made visible using the image analysis software ZEISS ZEN 3.2 (blue edition). To achieve uniformity in the approach for PMN count, in each H&E slide a consistent total area of interest was delineated at 2× magnification, excluding bone marrow, and then calculated in micrometer square (µm^2^) using the imaging software ZEISS ZEN 3.2 (blue edition). In each H&E slide, PMNs were then manually counted in 5 high-power (40× objective) magnification fields (HPFs) that were chosen randomly within the aforementioned delineated area [28,29]. The total PMN count for each H&E slide was then added from all 5 HPFs and divided by the preselected µm^2^ area to provide a count ratio (CR) [28,29]. The CR was then multiplied by 10^5^ to facilitate its histogram plotting.

### 2.6. Statistical Analysis

To compare bacterial load in the collected tissue at POD 3 and 13, bacterial count (CFU/mL) were converted to log10 then unpaired, two-tails *t*-test was used to compare means of the two groups. One-way ANOVA, Tuykey’s multiple comparisons test was used to compare the CR of PMNs between control and infected groups at POD 3 and 13. *p*-value < 0.05 was considered significant, with 95% confidence intervals (95% CI). Analyses were performed using GraphPad Prism 8.2.0. (GraphPad^TM^ Software, La Jolla, CA, USA).

## 3. Results

### 3.1. Establishing Infection

The establishment and progression of *S. aureus* infection was assessed using real-time imaging and plating for bacterial load. Figure 1A shows real-time imaging of bacterial replication within the infected hip joint using IVIS confirming that there was viable and persistent bacterial replication at the surgical site of all infected rats throughout the experimental timeline before they were euthanized. Quantification of the bacterial count in the periarticular soft tissue surrounding the infected prosthesis using CFUs confirmed the strong presence of bacterial infection at POD 3 and 13. The bacterial count at POD 13 was approximately a log lower than at POD 3 (Figure 1B).

### 3.2. Histological Analysis of Acetabulum

IHC analysis at POD 3 showed bacterial spread involving the soft tissue structures within the acetabulum, such as the fat-pad and the ligamentum teres (Figure 2). PMN recruitment to the infected tissues at POD 3 were identified by H&E (Figure 3). There was no evidence of *S. aureus* infiltration of the cartilage or subchondral bone. The MT stain showed early fibrosis involving the joint capsule, ligamentum teres, and fat-pad (Figure 3) compared to the non-infected controls (Appendix A).

At POD 13, IHC showed bacteria infiltrating the subchondral bone (Figure 2). The bacterial cells were also invading the capillary vessel walls within the infected tissue (Figure 2). Using H&E, PMNs were identified in the subchondral bone with evidence of sequestered bone fragments (Figure 3). The calculated CR of PMNs recruited by POD 13 was 4.48, which was significantly higher compared to POD 3 at 0.37 (*p* = 0.013), Figure 4. The MT stain showed increased fibrosis and tissue necrosis involving the cartilage and subchondral bone compared to POD 3 (Figure 3) and non-infected controls (Appendix A).

### 3.3. Histologic Analysis of Femur

At POD 3, IHC analysis showed infection to be contained within the intermedullary cavity of the femoral canal with no invasion of the surrounding cortical bone (Figure 5). The *S. aureus* cells were also invading the capillary vessel walls within the infected tissue of the femur (Figure 5).

PMN recruitment to the infected medullary cavity was identified by H&E (Figure 6). The MT stain only showed fibrosis involving the medullary canal (Figure 6). IHC analysis of bacterial infiltration at POD 13 showed bacteria infiltrating deeper within the cortical bone surrounding the infected medullary canal (Figure 5). Using H&E, PMNs can also be identified invading the outer and inner surfaces of the cortical bone with evidence of sequestered bone fragments (Figure 6). The CR of PMNs for POD 13 was 2.61, which was significantly higher compared to POD 3 at 0.69 (*p* = 0.022), Figure 4. The MT stain showed fibrosis and tissue necrosis involving the soft tissues and cortical bone to a much greater extent compared to what was seen at POD 3 (Figure 6) and compared to non-infected controls (Appendix A).

## 4. Discussion

*S. aureus* induced periprosthetic hip infections following arthroplasty is a unique and challenging clinical problem to manage. The incidence of PJI ranges from 1–2% in healthy individuals and up to 36% in immunocompromised patients [6,8,9]. The current standard therapy for PJI is very costly and has an alarming failure rate reaching 36% [7,30,31]. Treatment failure can lead to prolonged hospital admissions, poor quality of life for patients, significant morbidity, and may result in limb amputations or even death [10].

A critical step towards improving care for PJI patients is to develop clinically representative PJI models to guide the understanding of disease pathogenesis and to test the efficacy of various treatment strategies.

The first pilot total hip replacement in rats was performed in 1995 [32]. Since then, there has not been any documentation of other hip PJI models in small animals. Our group recently established and validated HA-PJI *S. aureus* model in rats using a cemented 3D printed titanium femoral stem [15,16]. This is the first published small in vivo model that resembles clinical and biomechanical features of the “ball in socket” design of a partial hip replacement. Using this model, we demonstrated the ability of bacteria to establish biofilm on the implant surface and to infect the surrounding bone and soft tissues mimicking clinical PJI. Our in vivo model also allows for real-time monitoring of live bacterial replication. Although our model is the first to mimic a PJI in a cemented ball in-socket prosthesis, it still required further histologic analysis to delineate its disease pathogenesis and guide the testing of future therapeutic interventions to manage HA-PJI.

Therefore, the current study was designed to fill this knowledge gap by performing histologic characterization of peri-implant tissue response to bacterial spread at early and late time points post bacterial infection. This histologic analysis provided the following information: (1) using IHC to map the pattern of bacterial spread in infected peri-implant tissues at early and late time points after infection; (2) assess differences in peri-implant tissue reaction, PMN infiltration, fibrosis, and necrosis at early and late time points of infection using H&E and MT stains. This analysis was performed in Sprague–Dawley rat after inoculating the surgical site with a clinical isolate of *S. aureus* and harvesting the femur and acetabulum at 3 days (early) and 13 days (late) post-surgery.

In our model, histological analysis demonstrated that *S. aureus* will infiltrate and persist in the bone at day 13 post-infection (summarized in Figure 7). As early as 3 days post-infection, histologic analysis of the femoral side showed bacteria to be mainly contained within the intermedullary canal but ultimately infiltrated peri-implant cortical bone at 13-days post-infection. IHC analysis also showed infiltration of bacteria within the acetabular tissues and eventually reaching subchondral bone at 13 days post-infection. This observation of *S. aureus’s* ability to infiltrate in the deep layers of the bone matrix causing osteomyelitis is critical to our understanding of the factors that might contribute to the challenges of curing PJI. It is of interest to note that *S. aureus* has the been reported to persist in PJI patients with rates of reinfection reaching up to 45% [33,34]. Bourget-Murray et al. recently published a retrospective cohort study reporting that DAIR with exchange of modular components performed in the context of HA-PJI was associated with poor chance of success [35,36]. These findings are in keeping with other recent publications in patients as well as in PJI in vivo models [37,38,39].

*S. aureus* capacity to persist and escape mechanical and immunological clearance could be attributed to several factors such *S. aureus* biofilm formation and survival within the host osteoblast, osteoclast, and immune cells [13,14,40,41]. Moreover, Bentley et al. provided preclinical evidence of *S. aureus* ability to invade and hide in the cortical bone and its canaliculi where they use it as their reservoir to cause chronic infections [25]. In alignment with this, our current histological analysis indicated the ability of *S. aureus* to invade the cortical bone at the later stage of the infection (i.e., 13 days post-infection) compared to early-stage infection at 3 days. However, we did not show *S. aureus* presence in the canaliculi of cortical bone.

Another key finding from our histological analysis was the irreversible tissue damage and fibrosis observed on the femoral and acetabular sides as early as 3 days post-infection. The tissue analyzed from the late-stage of infection at day 13 demonstrated a greater extent of fibrosis and necrosis involving the soft tissues and cortical bone of both femur, and acetabulum. Moreover, there was a significant increase in PMNs recruitment as the infection progressed. These histologic features highlight the development of osteomyelitis in our HA-PJI model at late-stage of *S. aureus* infection.

In Poultsides et al., an MRSA PJI rabbit model demonstrated evidence of osteomyelitis through an increase in PMNs, granulated and fibrotic tissue, nuclear debris, and necrosis as at 4 weeks of infection [42]. Moreover, Thompson et al. established a Pseudomonas aeruginosa PJI knee model in mice to assess in studying the pathogenesis of the Gram-negative infections [43]. The histological assessment of the bone/joint tissue detected inflammatory immune infiltration and other signs suggesting osteomyelitis at day 21 of infection. In a recently published study histologically examining bone tissue collected from osteomyelitis patients, an increase in neutrophil count, fibrosis, and bone necrosis were also demonstrated as features associated with osteomyelitis [27]. Our current study provided evidence of similar histological changes confirming the presence of osteomyelitis. Assessing the possible development of osteomyelitis as a part of the PJI disease process is an important step to assist in choosing the proper treatment approach.

Our current HA-PJI rat model is the first to establish a clinically representative animal model that mimics a functional cemented hip hemiarthroplasty implant. The strength of this study is that it demonstrated key histological features that mimics clinical findings for PJI associated osteomyelitis. The knowledge of the histopathological changes affecting the bone and joint tissues at early and late stages of infection in our rat PJI model is necessary to guide the utility of this model for testing future therapeutic strategies for hip HA-PJI.

This study was limited by the use of a small number of animals due its focus on qualitative histologic analysis. However, the effects observed in peri-implant infected tissues were quite significant at two selected time points post-infection. Another limitation of our study is examining the pathogenesis of a single PJI organism. We acknowledge that different microorganisms may show different patterns of tissue invasion and local tissue response. However, we focused on studying the effects of *S. aureus* in our model since it is the most prevalent causative microorganism responsible for PJI [44]. Therefore, we believe that our in vivo model can be used reliably for studying other aspects of disease pathogenesis caused PJI. Our future experiments will aim at utilizing the usage of our current 3D-printed titanium hip implants and the same surgical approach to establish a cemented HA-PJI model in rats having pertinent clinical comorbidities relevant to PJI (i.e., metabolic syndromes, diabetes). Moreover, our model could help in studying potential therapeutic effect of new treatment strategies [18] such as novel antibiotics, bacteriophages, and antimicrobial peptides, to name a few, in treating PJI and any associated potential histological changes.

## 5. Conclusions

This study expands the validity of a recently established cemented HA-PJI rat model. Histological analysis revealed significant quantitative and qualitative differences in peri-implant tissue response to *S. aureus* infection at early and late timepoints. Due to these common translational features, we believe that the current model can help assess novel interventions and technologies to treat PJI more accurately. In addition, this model can provide more insight into why certain surgical strategies such as DAIR may be associated with high failures rates in infected hip hemiarthroplasty, especially if utilized to treat later stages of infection.

## Figures and Tables

**Figure 1 microorganisms-12-01895-f001:**
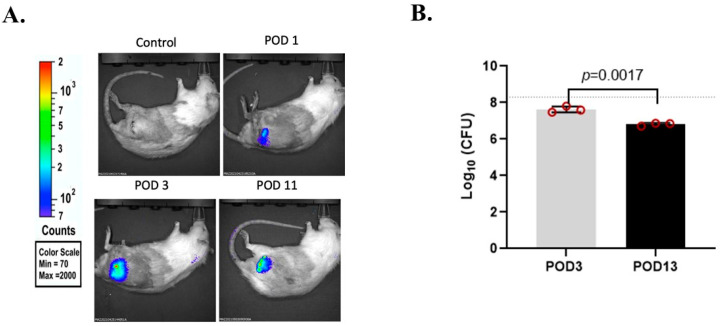
Establishing a persistent *S. aureus* infection. (**A**) Real-time in vivo imaging showing a luminescent signal at the site of the infected prosthesis, which confirms a persistent bacterial replication over POD 11-day. (**B**) Quantification of the bacterial load in the soft tissue surrounding the infected prosthesis confirmed the strong presence of bacterial infection at day 3 post-surgery (POD 3) and 13 (POD 13). Initial bacterial count (dotted line) was at log_10_ 8.3 (2 × 10^8^ CFU). *n* = 3 rats, mean ± SD.

**Figure 2 microorganisms-12-01895-f002:**
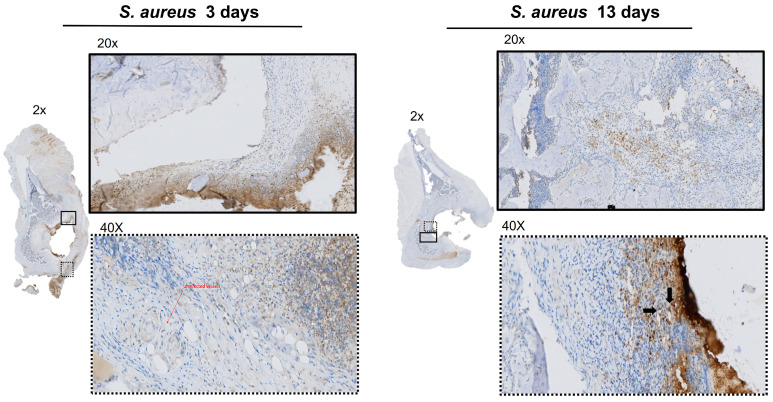
IHC staining of *S. aureus* in acetabulum. Rabbit polyclonal *S. aureus* antibody was used to localize bacterial presence within the collected tissue. Magnification at 2× and 20× show bacterial presence contained within the tissues of the capsule and fat pad of acetabulum at POD 3. On POD 13, bacteria infiltrates into the subchondral bone. The dash type of the box corresponds to the selected area of magnification. Magnification at 40× shows infection of capillary vessel walls at POD 13 compared to no infection at POD 3.

**Figure 3 microorganisms-12-01895-f003:**
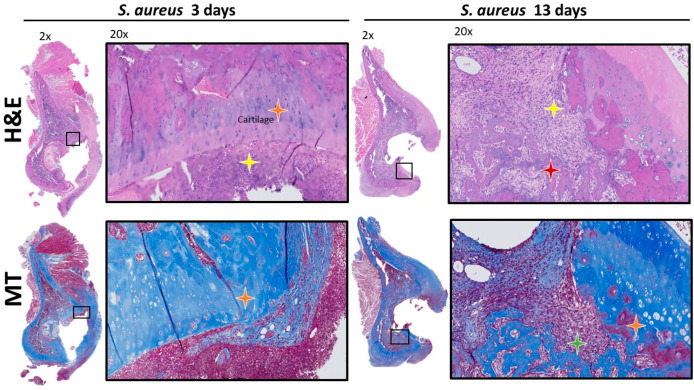
H&E and MT staining of acetabulum. H&E staining 2× and 20× shows increase PMN count and infiltration at POD 13 compared to POD 3 (yellow star). Red star shows bone sequestrum. Orange star shows intact cartilage. MT staining 2× and 20× shows fibrosis and fibroblast infiltration of cartilage and subchondral bone at POD 13 compared to POD 3. Red star shows bone sequestrum while green star shows fibroblasts. Orange star shows cartilage.

**Figure 4 microorganisms-12-01895-f004:**
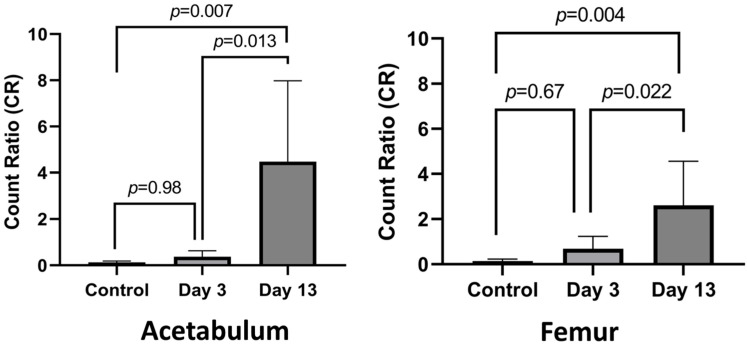
Polymorphonuclear (PMN) cell count ratio (CR). PMN CR in femur and acetabulum tissue is significantly higher at POD 13 compared to POD 3. *n* = 3 rats per cohort, 2 sections per rat, mean ± SD.

**Figure 5 microorganisms-12-01895-f005:**
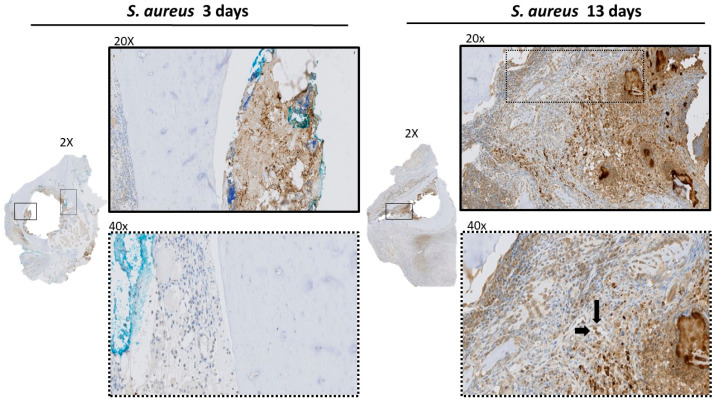
IHC staining of *S. aureus* in femur. Rabbit polyclonal *S. aureus* antibody was used to localize bacterial presence within femur tissue. Magnification at 2× and 20× show bacterial presence contained within the intramedullary canal of the femur at POD 3. At POD 13, bacteria infiltrates into the cortical bone. The dash type of the box corresponds to the selected area of magnification. Magnification at 40× shows infection of capillary vessel walls at POD 13 days compared to no infection at POD 3.

**Figure 6 microorganisms-12-01895-f006:**
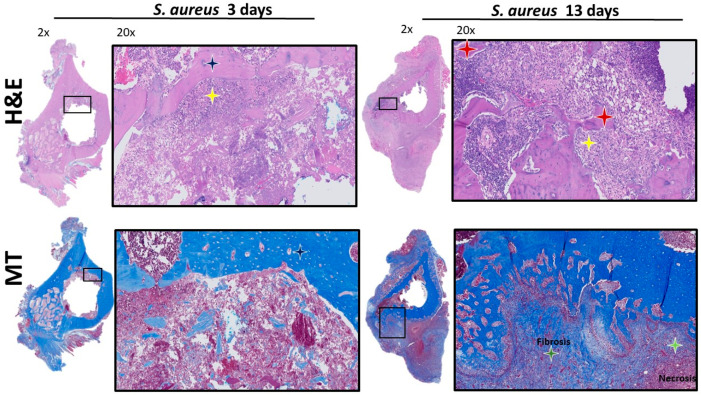
H&E and MT staining of femur. H&E staining 2× and 20× shows increase PMN count and infiltration at POD 13 compared to POD 3 (yellow star). Dark blue star (POD3) shows intact cortical bone while red star shows bone sequestrum. MT staining 2× and 20× shows fibrosis and necrosis of bone and surrounding muscle at POD 13 compared to POD 3. Light green star shows necrosis while dark green star represents fibrosis.

**Figure 7 microorganisms-12-01895-f007:**
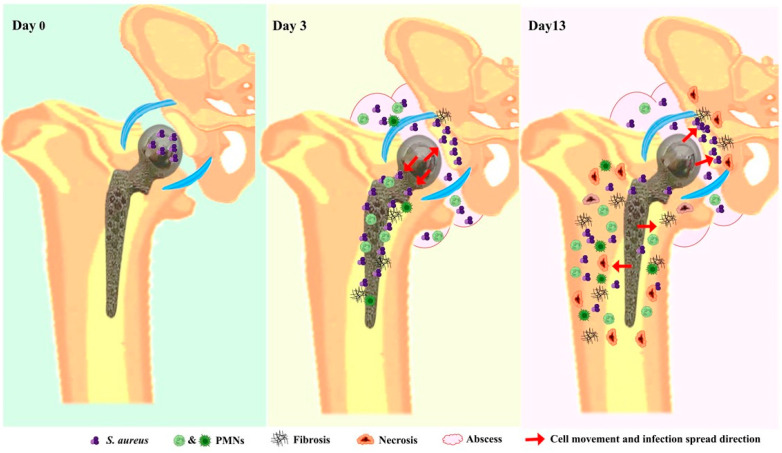
A schematic summarizing the histological findings at early and late stages of the bacterial infection in our PJI rat model. *S. aureus* was injected at the surgical site on the day of surgery (day 0). At day 3—post-surgery, an abscess formed, and *S. aureus* spread and persisted in the peri-implant tissues on the femoral and acetabular sides. Bacteria were largely contained within the intermedullary canal of the femur. Tissue analysis showed fibrosis, necrosis, and PMN recruitment. At day 13—post-surgery, *S. aureus* infiltrated peri-implant cortical bone and the subchondral bone of the acetabulum. This was associated with a greater extent of fibrosis and necrosis involving the soft tissues and cortical bone for femur and cartilage and subchondral bone of the acetabulum. Also, a significant increase in PMN count was detected.

## Data Availability

The original contributions presented in the study are included in the article/Appendix A, further inquiries can be directed to the corresponding author.

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
