# Peer review of "Mapping Staphylococcus aureus at Early and Late Stages of Infection in a Clinically Representative Hip Prosthetic Joint Infection Rat Model"

_microorganisms, 2024, doi:10.3390/microorganisms12091895_

Round 1

Reviewer 1 Report

Comments and Suggestions for Authors

Dear Authors, 

I express my sincere gratitude for the opportunity to review your manuscript. The effort of the authors is appreciated.

Upon reviewing your manuscript, several issues have raised my concerns:

-       Line 43-46 “Systemic antibiotic therapies often fail because they  are unable to reach minimum inhibitory concentrations (MIC) at the site(s) of infection due to the formation of biofilm on the implant surface and surrounding tissue, effectively shielding resident bacteria and leading to antibiotic tolerance” -- I believe that achieving the Minimum Inhibitory Concentration (MIC) in tissues, particularly in adjacent ones, primarily depends on the specific penetration of antibiotics into the tissues/site of infection and is subsequently related to the lack of access to microorganisms protected by biofilm.

-       Discussion

-       Line 272-“Periprosthetic hip infections following hip hemiarthroplasty (HA) is a unique and challenging clinical problem to manage where S. aureus is the main offending bacterium” -- Periprosthetic infections are equally important in all types of arthroplasties, not just hemiarthroplasties, regardless of etiology, and are not limited to Staphylococcus-related cases.

-       Line 276- “Treatment failure can lead to significant morbidity, and may result in limb amputations or even death” -- Other complications can also be taken into consideration.

-       Line 286-287- “Using this model, we demonstrated the ability of bacteria to establish biofilm on the implant surface and to infect the surrounding bone and soft tissues simiarly to clinical PJI” - The formation of biofilm on the surface of an implant was established many years ago, not just through this study. The extent of the periprosthetic infection, the inflammatory response, and their dynamics are the most important elements that you bring as new contributions.

A potential limitation of the study, or an area for further research, could be related to the administration of specific antistaphylococcal antibiotic therapy, in order to highlight its impact on the bacterial inoculum and on histopathological changes.

Information regarding whether the strain of Staphylococcus aureus was MRSA or not could also be included in the text.

Reviewer 2 Report

Comments and Suggestions for Authors

I would like to congratulate the authors on their original and innovative study.

There are a few points that I think could be improved:

- Summary: the items related to the results and conclusion could be more assertive.

Introduction: There is a need to contextualize the prosthesis and infection factor, failure rates, and the summary clinical importance of the model adopted. References need to be reviewed, there have been none in the last 2/3 years.

Methods: How long was the inoculum cultivated? It's not clear.

- Did the weight of the animals up to 700g and the 12-week lifespan try to mimic a clinical situation? If so, discuss.

- The methods need to be referenced.

Results: Figure 6 - I suggest making a graph with the result of the overall mean (SD) of the 3 animals per group.

Discussion: It's well described, but it's interesting to mention the differences between animal and human bone architecture. Recovery and combativeness. Above all, assess that it is not possible to extrapolate directly to the clinic. Other limitations could be mentioned, as well as the next studies that could be explored. 

References: Throughout the manuscript, it is necessary to review the citations, especially updating them.
